# Co-training and Co-distillation for
# Quality Improvement and Compression of Language Models

**Hayeon Lee**[1][*] **Rui Hou**[1] **Jongpil Kim**[1]
**Davis Liang**[2] **Hongbo Zhang**[1] **Sung Ju Hwang**[3] **Alexander Min**[1]
Meta AI[1]  Abridge AI[2]  KAIST[3]
hayeonlee@meta.com rayhou@meta.com jpkim@meta.com
davis@abridge.com hbzhang@meta.com sjhwang82@kaist.ac.kr alexmin@meta.com

## Abstract

Knowledge Distillation (KD) compresses computationally expensive pre-trained language models (PLMs) by transferring their knowledge to smaller models, allowing their use in resource-constrained or real-time settings. However, most smaller models fail to surpass the performance of the original larger model, resulting in sacrificing performance to improve inference speed. To address this issue, we propose Co-Training and Co-Distillation (CTCD), a novel framework that improves performance and inference speed together by co-training two models while mutually distilling knowledge. The CTCD framework successfully achieves this based on two significant findings: 1) Distilling knowledge from the smaller model to the larger model during co-training improves the performance of the larger model. 2) The enhanced performance of the larger model further boosts the performance of the smaller model. The CTCD framework shows promise as it can be combined with existing techniques like architecture design or data augmentation, replacing one-way KD methods, to achieve further performance improvement. Extensive ablation studies demonstrate the effectiveness of CTCD, and the small model distilled by CTCD outperforms the original larger model by a significant margin of 1.66 on the GLUE benchmark.

## 1 Introduction

In recent years, the high computational cost of pre-trained language models (PLMs) (Radford et al., 2019; Yang et al., 2019; Dai et al., 2019; Shoeybi et al., 2019; Li et al., 2020; Brown et al., 2020) become a constraint in serving resource-limited or real-time applications. Knowledge Distillation (KD) (Hinton et al., 2015; Romero et al., 2015) is a popular model compression technique to tackle this issue, with a smaller model (student) learning (distilling) from a larger model (teacher).

---

[*] Work done while interning at Meta AI.

The ideal scenario would involve compressing the teacher to match the size of the small student without any performance drop. However, despite intense research in the area of KD (Turc et al., 2019; Tsai et al., 2019; Tang et al., 2019; Jiao et al., 2020; Sanh et al., 2019a; Sun et al., 2019; Wang et al., 2020b,a), none of the existing methods have successfully avoided performance degradation during the distillation process. Some approaches have attempted to mitigate the performance gap by incorporating external factors. For instance, Jiao et al. (2020) incorporates data augmentation, while Wang et al. (2020b,a) focuses on designing student architecture. However, these approaches have limitations as they only provide supplementary techniques rather than addressing the fundamental issue of performance loss in KD. This raises an interesting question: Can we compress a model without scarifying the performance through KD?

In this work, we propose a novel framework called Co-Training and Co-Distillation (CTCD) for **improving the performance of a language model while compressing it** through KD. CTCD involves jointly training the teacher and the student models, allowing them to transfer knowledge to each other bidirectionally, from the teacher to the student and vice versa. Our work uncovers two key findings within the CTCD framework: Firstly, we demonstrate that transferring knowledge from the smaller model to the larger model during co-training significantly improves the performance of the larger model. In other words, by employing knowledge distillation (KD), we enhance the performance of the teacher model compared to standalone training. This is significantly different from conventional one-way KD, where the teacher model cannot benefit from the distillation process since it is no longer trained or distilled. Secondly, the improved performance of the larger model leads to further enhancements in the performance of the smaller model through KD from the teacher model to the student

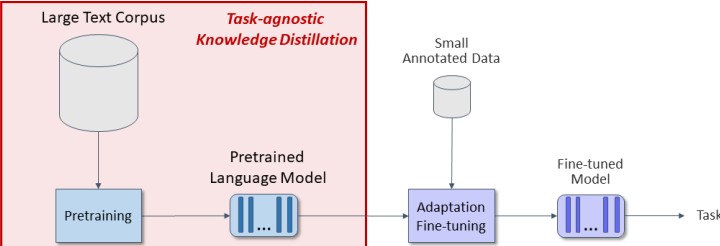

Figure 1: **Task-agnostic KD method** The proposed framework are task-agnostic KD methods that perform KD during the pre-training stage. This allows the distilled model to be deployed and fine-tuned on various downstream tasks.

model. These novel findings enable the smaller model to surpass the independently trained larger model while maintaining its inference efficiency within the CTCD framework. Moreover, the CTCD framework can be combined orthogonally with existing techniques for external factors such as student architecture design Wang et al. (2020b,a) or data augmentation (Jiao et al., 2020). We achieve this by replacing traditional one-way KD methods with the proposed CTCD framework, which holds promise for significant performance improvements in language modeling tasks.

In addition to our CTCD framework, we introduce a Community KD that utilizes CTCD framework with the pre-trained teacher model by combining CTCD with conventional KD. Community KD distills knowledge from two students to each other, as well as from the pre-trained teacher to each student, during the co-training of the students. We show that distilling from the other student as well as the pre-trained teacher is better than only one-way distillation from the pre-trained teacher, as is done in conventional KD. Note that the inference cost on a task is the same as in conventional KD, as we take one of the students distilled by Community KD when deploying it to the task.

We validate the effectiveness and efficiency of our CTCD on the GLUE benchmark (Wang et al., 2019), which contains different language understanding tasks. As shown in Figure 1, We focus on task-agnostic KD scenarios, where KD occurs during the pre-training stage. Then, we fine-tune the distilled PLMs on each downstream task and evaluate their performance. This is more challenging compared to task-specific KD, as it requires substantial computational resources to train PLMs on a large text corpus, and the distilled knowledge should be transferable across diverse downstream tasks. Our extensive ablation study revealed that the larger model benefits from the distillation of the smaller model, and the performance improvement

of the larger model further enhances the performance of the smaller model. In our experiments, the student model compressed by CTCD framework obtained 1.66 higher gain than the original large model trained using a stand-alone method, demonstrating that our approach can improve model quality and inference efficiency concurrently.

In summary, our contributions are as follows:

- We propose a novel knowledge distillation framework called Co-Training and Co-Distillation (CTCD) framework to improve the performance of models while compressing them through KD.

- Through our experiments, we demonstrate that distilling knowledge from the smaller model to the larger model during co-training improves the performance of the larger model.

- Additionally, we highlight that the enhanced performance of the larger model further boosts the performance of the smaller model, resulting in improved performance for both models.

- We provide valuable insights into adjusting loss weights and the length of the training phase for the effective application of the CTCD framework through extensive ablation study.

## 2   Related Work

**One-way Knowledge Distillation**   Knowledge distillation (KD) (Hinton et al., 2015) is a model compression technique in which a smaller student model distills knowledge from a larger, pre-trained teacher model. Recently, many researchers have explored KD to address a high computational complexity of a language model resulting from its increasing size. This includes task-agnostic KD methods for the pre-training stage (Sanh et al., 2019a; Wang et al., 2020b,a), task-specific KD method for

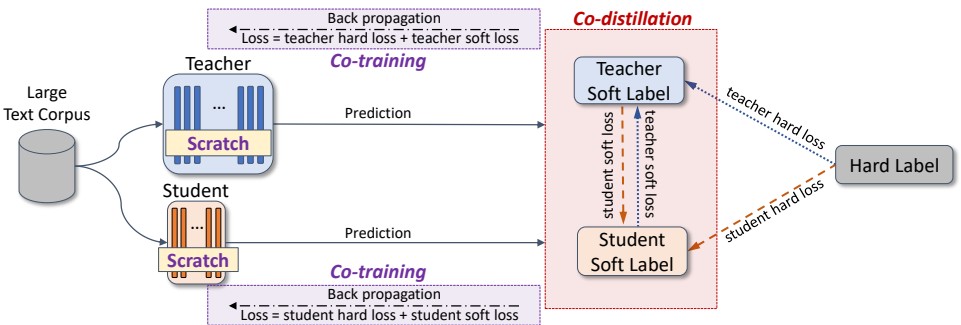

Figure 2: **Co-Training and Co-Distillation (CTCD)** During training, the teacher and student models can learn more effectively by comparing their prediction outputs not only against the ground truth but also against each other's predictions. We refer to the former as a "hard" loss and the latter as a "soft" loss. For example, the student soft loss captures the distance between the student's prediction and the teacher's prediction, and vice versa. This co-training and co-distillation approach improve the performance of the teacher model, which in turn benefits the performance of the student model.

the fine-tuning stage (Sun et al., 2019), and both of pre-training and fine-tuning stages (Jiao et al., 2020). Additionally, Wang et al. (2020b,a) have redesigned the architectures of student language models. However, the one-way KD can lead to a loss of knowledge, resulting in smaller models that generally have difficulty matching the performance of larger models, leading to performance degradation. In contrast, our CTCD can improve the quality of both teacher and student models, making it possible for the student to achieve the same quality as the original teacher model.

**Reversed Knowledge Distillation** Recently, researchers (Yuan et al., 2020; Qin et al., 2022; Lee et al., 2023) have demonstrated that reversed Knowledge Distillation (reversed KD), which transfers knowledge from a smaller or poorer model to a larger model, can improve the performance of the student model. In particular, Qin et al. (2022); Lee et al. (2023) investigated the application of reversed KD in the PLMs, showing that a larger model can benefit from a poorer and pre-trained model for a specific downstream task. Inspired by the success of reversed KD, we design a co-distillation framework that includes reversed KD to improve the performance of the teacher model by distilling knowledge from the smaller student model. Unlike existing reversed KD methods, which are limited to improving the performance of the larger model, our proposed co-distillation framework can achieve both performance improvement and model compression, by showing a better-quality teacher leads to a better-quality student.

## 3 Co-training and Co-distillation

We first introduce the concepts of co-training and co-distillation briefly:

**Co-training** trains two (different-sized) models (e.g., a teacher and student) concurrently with the goal of achieving similar model quality.

**Co-distillation** transfers knowledge in both directions between two models (e.g., a teacher and student), during co-training.

Figure 2 illustrates how co-trained models learn together by comparing their prediction outputs against predictions from each other and to the hard labels (or "ground truth"). We refer to the former as a "soft" loss and the latter as a "hard" loss. For instance, the soft loss of the student model measures the accuracy of the student's prediction by considering the teacher's prediction as a soft label, and vice versa.

**Task Formulation** Suppose that we are given a classification task with $K$ classes. For each training instance $x$ and its ground truth label $y$, we denote that the ground truth distribution over the labels is $q(k|x)$ ($q(k)$ for simplicity) where for each label $k \in \{1...K\}$, $q(y) = 1$ and $q(k) = 0$ for all $k \neq y$. For each $x$, the teacher model $t_\phi$ parameterized by $\phi$ and the student model $s_\theta$ parameterized by $\theta$ predict the probability of each label $k$ as $p_\phi^\tau(k|x)$ and $p_\theta^\tau(k|x)$, respectively as follows:

$$p_\phi^\tau(k|x) = f(\boldsymbol{z}^t) = \frac{\exp(z_k^t/\tau)}{\sum_{i=1}^{K} \exp(z_i^t/\tau)}$$

$$p_\theta^\tau(k|x) = f(\boldsymbol{z}^s) = \frac{\exp(z_k^s/\tau)}{\sum_{i=1}^{K} \exp(z_i^s/\tau)}$$

where $f$ is the softmax function, $\boldsymbol{z}^t = \{z_i^t\}_{i=1}^{K} = t_\phi(x)$ is the output logit of the teacher model, $\boldsymbol{z}^s = \{z_i^s\}_{i=1}^{K} = s_\theta(x)$ is the output logit of the student model, and $\tau$ is the temperature to soften $p_\phi(k)$ and $p_\theta(k)$.

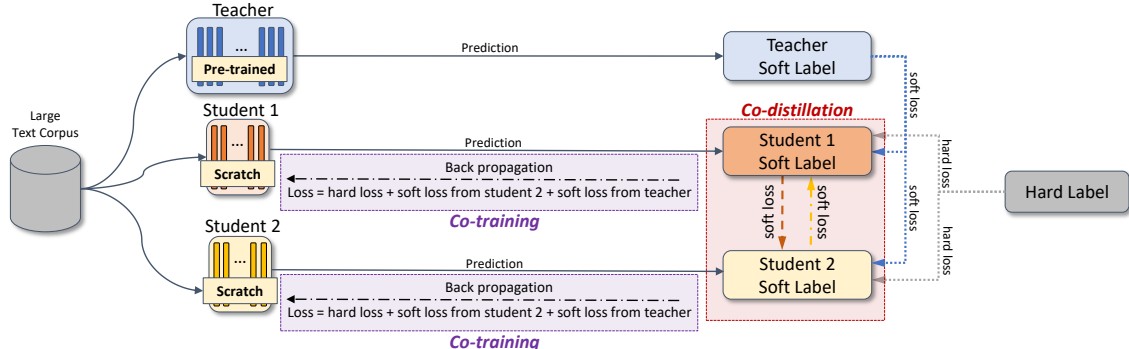

Figure 3: **Community KD** Different from conventional KD, where each student learns from the prediction of the pre-trained teacher only, the proposed approach learns each student from both prediction of the pre-trained teacher and prediction of another student during co-training. Note that since we take one of the pre-trained students to adapt it to downstream tasks, the inference cost is the same as the student training with the conventional KD.

The proposed objective $\mathcal{L}_{CTCD}(\boldsymbol{\theta}, \boldsymbol{\phi})$ consists of a normal KD objective $\mathcal{L}_{KD:t \to s}$ to distill knowledge from the teacher model to the student model and a reversed KD objective $\mathcal{L}_{ReKD:s \to t}$ to distill knowledge from the student model to the teacher model, during their co-training.

**Teacher → Student** The normal KD objective $\mathcal{L}_{KD:t \to s}(\boldsymbol{\theta}, \boldsymbol{\phi})$ aims to train the student model by minimizing a weighted sum of the cross-entropy loss $H(q, p_{\boldsymbol{\theta}})$ between the ground truth $q$ and student prediction $p_{\boldsymbol{\theta}}$ and Kullback-Leibler divergence (KL divergence) $D(p_{\boldsymbol{\phi}}^{\tau}, p_{\boldsymbol{\theta}}^{\tau})$ between the predictions of the student and the teacher as follows:

$$\mathcal{L}_{KD}(\boldsymbol{\theta}, \boldsymbol{\phi}) = \alpha_h \cdot H(q, p_{\boldsymbol{\theta}}) + \alpha_s \cdot D(p_{\boldsymbol{\phi}}^{\tau}, p_{\boldsymbol{\theta}}^{\tau}) \tag{1}$$

where

$$H(q, p_{\boldsymbol{\theta}}) = -\sum_{k=1}^{K} q(k) \log(p_{\boldsymbol{\theta}}(k)),$$

$$D(p_{\boldsymbol{\phi}}^{\tau}, p_{\boldsymbol{\theta}}^{\tau}) = \sum_{k=1}^{K} p_{\boldsymbol{\phi}}^{\tau}(k) \cdot \log \frac{p_{\boldsymbol{\phi}}^{\tau}(k)}{p_{\boldsymbol{\theta}}^{\tau}(k)},$$

$\alpha_h$ and $\alpha_s$ are weighting hyper-parameter values for the cross-entropy loss and KL divergence, respectively. We regard the cross-entropy loss $H(q, p_{\boldsymbol{\theta}})$ as the hard loss for the student model, KL divergence $D(p_{\boldsymbol{\phi}}^{\tau}, p_{\boldsymbol{\theta}}^{\tau})$ as the soft loss for the student model, and following BERT (Devlin et al., 2019), $H(q, p_{\boldsymbol{\theta}})$ denotes the Masked Language Modeling loss (MLM loss). In the KD objective, we consider the teacher parameters $\boldsymbol{\phi}$ as constant since we only train the student parameters $\boldsymbol{\theta}$ while the teacher model $t_{\boldsymbol{\phi}}$ is fixed:

$$\mathcal{L}_{KD:t \to s}(\boldsymbol{\theta}, \text{StopG}(\boldsymbol{\phi})) \tag{2}$$

where $\text{StopG}(x)$ denotes that we do not compute the gradient of $x$. In the conventional KD method, Equation (2) is the final objective to learn the student model only with the pre-trained teacher model.

**Student → Teacher** Different from such a one-way KD method, we introduce the reversed KD objective $\mathcal{L}_{ReKD:s \to t}(\text{StopG}(\boldsymbol{\theta}), \boldsymbol{\phi})$ to train the teacher model $t_{\boldsymbol{\phi}}$ as follows:

$$\mathcal{L}_{ReKD:s \to t}(\text{StopG}(\boldsymbol{\theta}), \boldsymbol{\phi}) = \\ \beta_h \cdot H(q, p_{\boldsymbol{\phi}}) + \beta_s \cdot D(p_{\boldsymbol{\theta}}^{\tau}, p_{\boldsymbol{\phi}}^{\tau}) \tag{3}$$

where $\beta_h$ and $\beta_s$ are weighting hyper-parameter values of the hard loss $H(q, p_{\boldsymbol{\phi}})$ and soft loss $D(p_{\boldsymbol{\theta}}^{\tau}, p_{\boldsymbol{\phi}}^{\tau})$ for the teacher model, respectively. By minimizing KL divergence $D(p_{\boldsymbol{\theta}}^{\tau}, p_{\boldsymbol{\phi}}^{\tau})$ between the predictions of the student model ($p_{\boldsymbol{\theta}}^{\tau}$) and the teacher model ($p_{\boldsymbol{\phi}}^{\tau}$), the teacher model learns from the student model. In the reversed KD objective, we only train the teacher model by applying $\text{StopG}(x)$ to the gradient of the student parameters $\boldsymbol{\theta}$.

**Co-training** With the Equations (2) and (3), we get the final objective $\mathcal{L}_{CTCD}(\boldsymbol{\theta}, \boldsymbol{\phi})$ as follows:

$$\boldsymbol{\theta}^*, \boldsymbol{\phi}^* = \underset{\boldsymbol{\theta}, \boldsymbol{\phi}}{\operatorname{argmin}} \, \mathcal{L}_{CTCD}(\boldsymbol{\theta}, \boldsymbol{\phi}) = \\ \mathcal{L}_{KD}(\boldsymbol{\theta}, \text{StopG}(\boldsymbol{\phi})) + \mathcal{L}_{ReKD}(\text{StopG}(\boldsymbol{\theta}), \boldsymbol{\phi}) \tag{4}$$

**Adapting to Downstream Task** After model co-training/-distillation, the trained smaller (student) model $s_{\boldsymbol{\theta}^*}$ with trained parameter $\boldsymbol{\theta}^*$ can be deployed for multiple downstream tasks to improve inference efficiency. To fine-tune the model for a specific downstream task, we adapt the trained parameter $\boldsymbol{\theta}^*$ using the dataset for that task.

# 4 Community KD

Furthermore, we introduce an advanced CTCD application named Community KD that can utilize CTCD framework with the pre-trained teacher model, as shown in Figure 3. It consists of a pre-trained teacher $t_{\phi^*}$ with pre-trained parameters $\phi^*$ and two students $s_{\theta_1}$ and $s_{\theta_2}$ parameterized by $\theta_1$ and $\theta_2$, respectively. During the co-training of two students, each student learns from the hard labels, soft labels generated from the pre-trained teacher predictions, and soft labels generated from other student predictions. In other words, we conduct one-way knowledge distillation from the pre-trained teacher to each student by minimizing KL divergence between the teacher prediction and predictions of each student $D(p_{\theta_1}^\tau, p_{\phi^*}^\tau)$ and $D(p_{\theta_2}^\tau, p_{\phi^*}^\tau)$ and co-distillation between students in both directions by minimizing $\mathcal{L}_{CTCD}(\theta_1, \theta_2)$. The final objective $\mathcal{L}_{CM}(\theta_1, \theta_2, \text{StopG}(\phi^*))$ is as follows:

$$\theta_1^*, \theta_2^* = \operatorname*{argmin}_{\theta_1, \theta_2} \mathcal{L}_{CM}(\theta_1, \theta_2, \text{StopG}(\phi^*)) =$$
$$\mathcal{L}_{CTCD}(\theta_1, \theta_2) + D(p_{\theta_1}^\tau, p_{\phi^*}^\tau) + D(p_{\theta_2}^\tau, p_{\phi^*}^\tau) \tag{5}$$

We select **one** of the two students distilled by Community KD ($\theta^* = \theta_1^*$ or $\theta^* = \theta_2^*$) and fine-tune the selected single student $s_{\theta^*}$ for downstream tasks, resulting that the inference cost does not increase compared with the conventional KD method.

# 5 Experiment

We present a comprehensive analysis of the proposed CTCD method through empirical experiments. In Section 5.1, we validate our CTCD method by comparing the performance of small models distilled by CTCD to the original large model on the GLUE benchmark (Wang et al., 2019). In Section 5.2, we analyze the impact of co-distillation by adjusting loss weights for the soft losses of a student and a teacher. In Section 5.3, we study the impact of training length on CTCD method, allowing us to determine the optimal training length for CTCD method. In Section 5.4, we demonstrate the efficacy of Community KD by comparing the performance of a model distilled by the Community KD to a model distilled by the one-way KD.

**Implementation details**   We use a learning rate of 5e-4, linear warm-up of 5%, AdamW optimizer (Loshchilov and Hutter, 2019), and batch size of 128 with A100 GPUs for pre-training. We train the teacher and student models from scratch

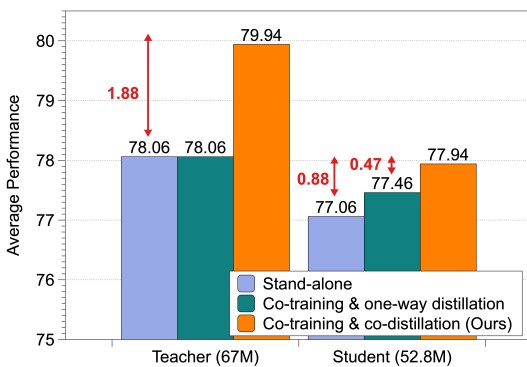

Figure 4: **Average performance on dev sets of GLUE benchmark after training 10 epochs** Learning from the student improves the performance of the teacher by an average of 1.88. Such improvement in teacher performance leads to improvement in student quality from 77.46 to 77.94.

for 20 epochs in Section 5.1. To analyze the effectiveness of CTCD method, we train the teacher and student models for 10 epochs and 20 epochs in Section 5.2 and Section 5.3, respectively. In Section 5.4, we train the models for 3 epochs after parameter remapping, which is the same as in conventional one-way KD method (Sanh et al., 2019a).

**Training Time**   For the one-way distillation, we need 1 GPU day to train the teacher for 10 epochs and 1.3 GPU days to distill knowledge from the pre-trained teacher to the student for another 10 epochs, which consumes a total of 2.3 days. For CTCD, it takes 3 GPU days to train both teacher and student models from scratch for 10 epochs. We use the automatic mixed precision (AMP) of PyTorch (Paszke et al., 2019) to accelerate training for all our models.

**Dataset**   To validate our CTCD method, we use a reduced dataset (30M) created by uniformly sampling 1 out of every 4 sentences from the original pre-training dataset (BookCorpus (Zhu et al., 2015) + Wikipedia (Foundation)) used in the conventional one-way KD method (Sanh et al., 2019a). We evaluate our distilled models on the dev sets of the GLUE benchmark (Wang et al., 2019), which consists of nine sentence-level classification tasks. In Section 5.4, we use the original pre-training dataset to train Community KD method.

**Model Architecture**   We use a 6-layer BERT (Devlin et al., 2019) model as the teacher and a 4-layer BERT model as the student to analyze the effectiveness and efficiency of our CTCD method. In Section 5.4, we use a pre-trained BERT-base model as the teacher and a 6-layer BERT model as the student.

|  |  | Performance | GAP w/ Teacher |
|---|---|---|---|
| Original Teacher |  | 78.06 | - |
| Student | One-way Distil. (10 epoch) | 77.46 | -0.60 |
|  | **CTCD** (10 epoch) | **77.94** | **-0.12** |
|  | One-way Distil. (20 epoch) | 78.39 | +0.33 |
|  | **CTCD** (20 epoch) | **79.12** | **+1.66** |

Table 1: **Average performance on dev sets of GLUE benchmark** The student distilled by CTCD significantly outperforms the original teacher trained using the stand-alone method, achieving a higher gain of **1.66**.

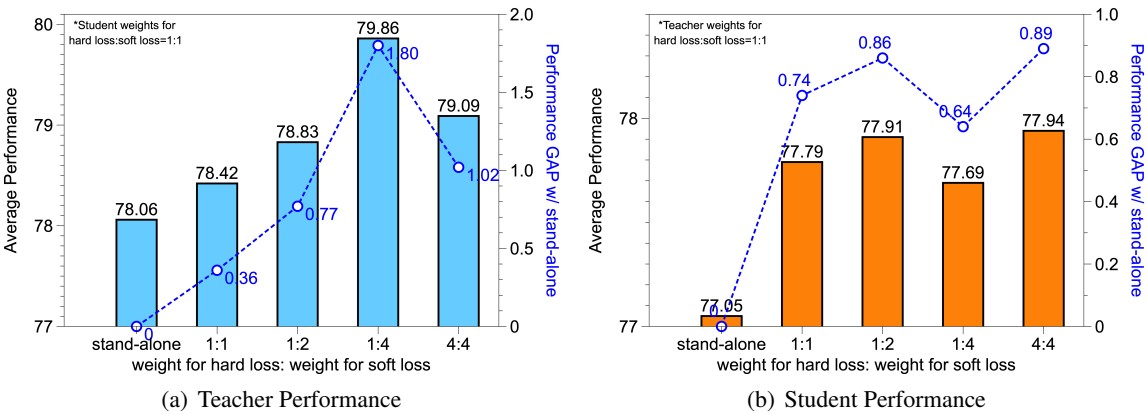

(a) Teacher Performance       (b) Student Performance

Figure 5: **Adjusting Loss Weight** We investigate the impact of the distillation for the teacher model and student model by adjusting loss weights ($\alpha_h, \alpha_s, \beta_h, \beta_s$) for hard loss and soft loss. (a) We (co-)train the teacher model distilling knowledge from the student by fixing $\alpha_h : \alpha_s = 1 : 1$ and varying $\beta_h : \beta_s$ on the large text corpus. (b) We (co-)train the student model distilling knowledge from the teacher by fixing $\beta_h : \beta_s = 1 : 1$ and varying $\alpha_h : \alpha_s$ on the large text corpus. Then we report the average performance of each pre-trained model after fine-tuning it on downstream tasks (dev sets) of GLUE benchmark.

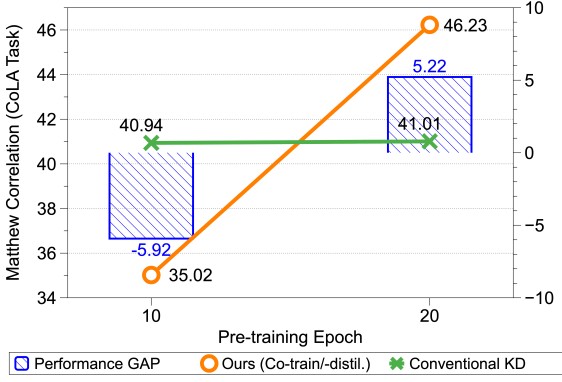

Figure 6: **Length of Training** We pre-trained student models under two different training lengths 10/20 epochs while distilling knowledge from teacher models via ours or the conventional KD method. Then we adapt pre-trained student models on CoLA task. With enough longer training (20 epoch), the student model distilled by ours significantly outperforms the student model distilled by the conventional KD method, with a higher gain of 5.22.

## 5.1 Could Knowledge Distillation Help Improve Performance?

In Table 1 and Figure 4, we show the average performance of models trained using different methods on a large text corpus and fine-tuned against the GLUE benchmark. **Stand-alone** trains a model

without using any knowledge distillation. **Co-training & One-way distillation** trains teacher and student models together from scratch, with knowledge only flowing from the teacher to the student. **Co-training & Co-distillation (Ours)** is CTCD method, which trains both teacher and student models together from scratch and distills knowledge between each other in both directions. For distillation methods, we set the weights of the hard losses for the teacher and student to 1. The weights of the soft losses are chosen from the set $\{0.5, 1, 2, 4\}$, and the results are reported with the best-performing weights.

**1) Overall Results** As shown in Table 1, the student distilled by CTCD method significantly **outperforms the original teacher** trained using the stand-alone method on the average performance of the GLUE benchmark, achieving a higher gain of **1.66**. Furthermore, the student model distilled by our CTCD method outperforms the student distilled by the one-way distillation method on the average performance of the GLUE benchmark. After training for 10 and 20 epochs, the student distilled by our CTCD method consistently has higher gains

than the student distilled by one-way distillation, as 77.46 vs. **77.94** and 78.39 vs. **79.12**, respectively.

**2) Does learning from a small and weak student provide a performance benefit to the teacher?** As shown in Figure 4, the distillation of knowledge from the student model to the teacher model has been shown to significantly improve the quality of the teacher model, with an average increase of 1.88 compared to teacher training methods that do not incorporate such distillation process (such as Stand-alone and Co-training & one-way distillation).

**3) Is the teacher's performance improvement reflected in the student's performance improvement?** Furthermore, we find that a better teacher leads to further performance improvement of the student from 77.46 to 77.94. The results demonstrate that the distillation process successfully improves the performance of both the teacher and student. The student trained with our CTCD method achieves better performance than students trained with the Stand-alone or Co-training & one-way distillation methods, with an average improvement of 0.88 and 0.47, respectively.

### 5.2 In-depth Study 1: Adjusting Loss Weights

In this Section, we investigate the impact of distillation on the performance of both the student and the teacher by adjusting loss weights $(\alpha_h, \alpha_s, \beta_h, \beta_s)$ for hard loss and soft loss. For example, setting $\alpha_h : \alpha_s = 1 : 4$ emphasizes learning from the teacher's knowledge (i.e., the soft loss for the student) $4\times$ more than the ground truth label distribution (i.e., the hard loss for the student), during co-training. This allows us to better understand the effect of distillation on each model and optimize their performance.

**Teacher side** In Figure 5(a), we co-train the teacher model using distillation to transfer knowledge from the student model. We fix the weighting values for the losses of the student to $\alpha_h : \alpha_s = 1 : 1$ and vary the weighting values for the losses of the teacher, $\beta_h : \beta_s$, while training on a large text corpus. We then evaluate the average performance of the pre-trained teacher models on downstream tasks from the dev sets of GLUE benchmark.

Our results show that co-training the teacher with distillation outperforms training the teacher alone, regardless of the weighting values for the soft loss, by obtaining higher gains of 0.36, 0.77, 1.80, and 1.02 for $\beta_h : \beta_s = 1 : 1, 1 : 2, 1 : 4$, and $4 : 4$, respectively. Additionally, we find that giving more

weight to the soft loss of the teacher during training $(\alpha_h : \alpha_s : \beta_h : \beta_\mathbf{s} = 1 : 1 : 1 : \mathbf{1} \rightarrow 1 : 1 : 1 : \mathbf{2} \rightarrow 1 : 1 : 1 : \mathbf{4})$ leads to improved performance, with an average score of $78.42 \rightarrow 78.83 \rightarrow 79.86$. Furthermore, we observe that emphasizing only the soft loss of the teacher $(1 : 1 : 1 : 4)$ yields better performance than emphasizing both the hard and soft losses of the teacher $(1 : 1 : 4 : 4)$, with an average score of 79.86 vs. 79.09.

**Student side** We find that the student model's performance is not sensitive to the weighting values for the hard and soft losses of the student, $(\alpha_h : \alpha_s)$. Regardless of the chosen values, co-training the student with distillation consistently improves its performance compared to training the student alone. For instance, when we emphasize the soft loss of the student by increasing the weighting value for $(\alpha_s)$ as $1 : \mathbf{1} : 1 : 1 \rightarrow 1 : \mathbf{2} : 1 : 1 \rightarrow 1 : \mathbf{4} : 1 : 1$, we observe similar levels of performance for the student model.

### 5.3 In-depth Study 2: Length of Training

We studied the impact of co-training length on the effectiveness of the CTCD method (see Figure 6). We find that longer training leads to improved performance, as demonstrated by our experiments using two different training lengths: 10 epochs and 20 epochs. After pre-training the student models with these different lengths, we adapted them to the CoLA downstream tasks and evaluated their performance using Matthew Correlation.

**Results** By increasing the (co-)training length from 10 epochs to 20 epochs, CTCD (Ours) significantly improves the performance of the student model from 35.02 to **46.23**, with a gain of 11.21. This outperforms the conventional KD method, which only achieves a gain of 0.07 from 40.94 to **41.01**, despite a longer training time. The conventional KD method relies on a pre-trained teacher model to train the student model, which allows for fast convergence but limits the learning of the student model. In contrast, the CTCD method allows for additional performance gains for both the teacher and student models by enabling them to learn and grow together during co-training. This can provide further benefits to the student model's performance with longer co-training.

### 5.4 Efficacy of Community KD

We compare the proposed Community KD with the conventional one-way KD method (Sanh et al.,

| Downstream Task | | MNLI | QQP | QNLI | SST-2 | CoLA | STSB | | MRPC | | RTE | Average |
| Metric | AMP | Acc. | Acc. | Acc. | Acc. | Matthew. | Pearson. | Spear. | F1 | Acc. | Acc. | Acc. |
| Dataset Size | | 392.7k | 363.8k | 104.7k | 67.3k | 8.5k | 5.7k | | 3.7k | | 2.5k | |
|---|---|---|---|---|---|---|---|---|---|---|---|---|
| **Teacher** BERT (109M) | | 84.17 | 90.89 | 90.68 | 91.86 | 57.54 | 88.84 | 88.56 | 89.31 | 85.04 | 65.34 | 83.23 |
| **Student (67M)** One-way KD (Sanh et al., 2019b) | FP32 | **81.93** | 90.05 | 87.72 | 90.94 | 52.03 | 86.28 | 86.07 | 87.94 | 82.59 | 57.76 | 80.33 |
| **Ours: Student 1** | FP16 | 81.88 | **90.17** | 88.24 | **91.51** | 54.82 | **86.70** | **86.49** | 89.76 | **85.29** | **59.21** | **81.40** |
| **Ours: Student 2** | FP16 | 81.34 | 89.75 | **88.37** | 90.71 | **56.08** | 86.42 | 86.44 | **89.80** | **85.29** | 59.20 | 81.34 |

Table 2: **Efficacy of Community KD** The pre-trained BERT and 6-layer BERT is the teacher model and student architecture, respectively, for both ours and the conventional one-way KD method. We fine-tune the distilled students on dev sets of GLUE benchmark. We observe that learning from the soft knowledge of different student model improves performance over the conventional one-way KD method on most downstream tasks.

2019a). To ensure a fair comparison, we use the pre-trained BERT-base as the teacher model for both methods and the 6-layer BERT as the student, which is the same architecture used in the conventional one-way KD method. As described in Section 4, we train two student models concurrently and they learn from the pre-trained BERT, the ground truth labels, and each other's knowledge. Note that since we fine-tune **one** of the two students distilled by Community KD for downstream tasks, the inference cost is the same as the conventional one-way KD method. In Table 2, we report the results of BERT and the conventional one-way KD method using checkpoints provided by Hugging Face (HuggingFace) and both students (Ours: Student 1 and Ours: Student 2) on the dev sets of the GLUE benchmark. We apply Automatic Mixed Precision (AMP) (Paszke et al., 2019) to Community KD, which typically speeds up training but may hurt performance.

**Results** The results presented in Table 2 shows that Community KD, leads to improved performance on downstream tasks such as QQP, QNLI, SST-2, CoLA, STSB, MRPC, and RTE, even when applying quantization techniques. Specifically, the average performance gain of the student model distilled using our Community KD method is 1.04 (1.2%) higher than that of the student model distilled by the conventional one-way KD method. This suggests that incorporating knowledge distillation from both a student model and a pre-trained teacher model is more effective than only using knowledge distillation from the pre-trained teacher model.

## 6   Limitations & Future Work

**Limitations** The proposed method co-train models from scratch and may require a longer pre-training time than the conventional KD method. However, as we described in Section 5.3, when the student model is trained long enough with its

teacher, it can outperform the models trained with the conventional KD on the downstream task. The proposed co-training method may increase the overall training cost compared with one-way distillation, and it may become a performance bottleneck depending on training resource constraints. However, note that CTCD can improve model quality while having the same inference cost as the one-way distillation on downstream tasks.

**Future Work**   **1) Architecture Sharing.** Models can share some of their architectures by reusing the output of such architectures and updating them together during back-propagation. This may help reduce the additional computing and memory overhead incurred by model co-training, while improving the model quality, especially for the student model. **2) Integration of Student Architecture Design and Data Augmentation.** Future research can focus on effectively combining the CTCD framework with student architecture design and data augmentation techniques. This integration provides a promising alternative to traditional one-way knowledge distillation methods, leading to significant improvements in language modeling tasks.

## 7   Conclusion

The size and complexity of pre-trained language models (PLMs) can hinder their practicality for online downstream tasks. To address this, we introduced a novel framework called co-training and co-distillation (CTCD). By training models of different sizes together and extracting inter-model knowledge in both directions, the proposed CTCD framework improves both model efficiency and performance. The proposed framework overcomes the trade-off between efficiency and performance in traditional one-way knowledge distillation methods. Notably, our compressed models achieved an impressive gain of 1.66 on the GLUE benchmark, outperforming large models trained using standalone methods.

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
