# OpenReview forum: "Co-training and Co-distillation for Quality Improvement and Compression of Language Models"
_EMNLP/2023/Conference — EMNLP 2023 Findings_

### Official Review · Reviewer_kfJK · 2023-07-31

**Soundness:** 3

**Excitement:**

3: Ambivalent: It has merits (e.g., it reports state-of-the-art results, the idea is nice), but there are key weaknesses (e.g., it describes incremental work), and it can significantly benefit from another round of revision. However, I won't object to accepting it if my co-reviewers champion it.

**Paper Topic And Main Contributions:**

Knowledge Distillation (KD) compresses large pre-trained language models by transferring their knowledge to smaller models, enabling use in resource-constrained settings, but often sacrifices performance. To address this, the authors propose Co-Training and Co-Distillation (CTCD), a framework that co-trains two models of different sizes, mutually distilling knowledge between them. CTCD improves both model performance and inference speed, based on findings that: 1) Distilling knowledge from the smaller to the larger model boosts the larger model's performance, and 2) The improved larger model further enhances the smaller model's performance. CTCD shows promise to combine with techniques like architecture design and data augmentation, replacing one-way KD, for further gains. Extensive ablation studies demonstrate CTCD's effectiveness, where the small model distilled by CTCD significantly outperforms the original larger model by 1.66 on GLUE benchmark.

**Reasons To Accept:**

1. This paper addresses a critical limitation of knowledge distillation (KD) methods - the tradeoff between model compression and performance degradation.
2. This paper proposes a novel co-training and bi-directional distillation approach. Most prior work uses one-way distillation which loses knowledge. In contrast, CTCD allows mutual transfer between teacher and student models.
3. Experimental results indicate two important empirical findings related to co-distillation: 1) Distilling knowledge from smaller to larger model boosts the larger model's performance. 2) The improved larger model further enhances the smaller model's performance. Additionally, the distilled small model significantly outperforms the original large model by 1.66 on GLUE benchmark.

**Reasons To Reject:**

1. My major concern is about the source of the performance improvement, especially the finding that distilling knowledge from smaller to larger models boosts the larger model's performance. I am wondering if the improvement could be attributed to the values of the hyperparameters α_h, α_s, β_h and β_s. Although the training epoch is 10 for all different settings, larger values for these hyperparameters result in longer training. I would be convinced if the authors could provide more empirical analysis, e.g. ablation studies or theoretical analysis, to demonstrate that the performance gains are due to the distillation approach rather than the hyperparameter settings.

2. Another key concern is the selection of teacher and student models. The authors mentioned using 6-layer BERT as the teacher and 4-layer BERT as the student. This seems like an unusual configuration. A more reasonable setting would be to use 12-layer BERT as the teacher and 6-layer or 4-layer BERT as the student model.

**Reproducibility:**

4: Could mostly reproduce the results, but there may be some variation because of sample variance or minor variations in their interpretation of the protocol or method.

**Reviewer Confidence:**

4: Quite sure. I tried to check the important points carefully. It's unlikely, though conceivable, that I missed something that should affect my ratings.

---

> ### Author Rebuttal · Authors · 2023-08-28
>
> We thank you for your positive feedback that our work **addresses a critical limitation of KD**, is **novel** and shows two **important** empirical findings.
>
> ***
> **1. I would be convinced if the authors could provide empirical analysis to demonstrate that the performance gains of the larger model are attributed to the knowledge distillation approach rather than the hyperparameter settings.**
> - Thank you for your insightful comment, and we agree that demonstrating that the performance improvement of a large model is due to the knowledge distillation process is important.
> - Fortunately, as demonstrated in **Figure 5** and **Section 5.2** of the main paper, we have already conducted experiments addressing the main concern of the reviewer. We empirically analyzed the impact of knowledge distillation from the small model to the large model based on the sizes of hyperparameter values. We observed that, **regardless of the hyperparameter sizes, knowledge distillation from the small to the larger model consistently enhances the performance of the large model**. In other words, we empirically verified that **the performance improvement of the large model is attributed to the effect of knowledge distillation, rather than the hyperparameter settings**.
> - Specifically, as shown in **Figure 5 (a)** of the main paper or the below **Table**, we investigate the effectiveness of distilling knowledge from small to large models by (co-)training the large model distilling knowledge from the small model. To this end, we fix the hyperparameters for the hard and soft (distillation) losses of the student to $\alpha_h:\alpha_s = 1:1$ and vary the values for the hard and soft (distillation) losses of the teacher, $\beta_h:\beta_s$ while training on a large text corpus. We then evaluate the average performance of the pre-trained teacher models on downstream tasks from the GLUE benchmark.
>
> |         | Stand-alone (w/o KD) |  $\beta_h:\beta_s=1:1$  |  $\beta_h:\beta_s=1:2$  |  $\beta_h:\beta_s=1:4$  |  $\beta_h:\beta_s=4:4$  |
> |:-------:|:--------------------:|:-----:|:-----:|:-----:|:-----:|
> | Average |         78.06        | 78.42 | 78.83 | 79.86 | 79.09 |
> |   Gain  |           -          | +0.36 | +0.77 | +1.80 | +1.02 |
>
> - Our results demonstrate that the distilled large model achieves significantly greater performance improvements of 0.36, 0.77, 1.80, and 1.02, respectively, compared to the independently trained large model (Stand-alone) with a performance of 78.06, for **all cases** of $\beta_h:\beta_s = 1:1, 1:2, 1:4$, and $4:4$, consistently. (For a more detailed explanation, please refer to Section 5.2.)
> - Interestingly, we additionally observe experimental results that partially support the reviewer’s argument: larger hyperparameter values ($\beta_s$) for soft losses can lead to longer training effects. Specifically, giving more weight to the soft loss of the larger model during training ($\alpha_h:\alpha_s:\beta_h:\mathbf{\beta_s} = 1:1:1:\mathbf{1} \rightarrow 1:1:1:\mathbf{2} \rightarrow 1:1:1:\mathbf{4}$) brings about performance improvement. The average scores increased from $78.42$ to $78.83$, and further to $79.86$. However, we observe that a larger hyperparameter value for the hard loss ($\beta_h$) does not necessarily lead to a longer training effect. Emphasizing only the soft loss of the teacher (1:1:1:4) actually results in better performance compared to emphasizing both the hard and soft losses of the teacher (1:1:4:4), with average scores of 79.86 vs. 79.09.
> - We believe that this finding is a meaningful analysis that contributes to a better comprehension of the proposed method and further strengthens the effectiveness of our method. In other words, knowledge distillation from the small model to the large model brings performance improvements ranging from 0.36 to 1.80 in all cases when compared to independently training the larger model, due to the effect of knowledge distillation. Furthermore, increasing the hyperparameter values for soft losses ($\beta_s$) allows for reducing the training time of the larger model or achieving a greater distillation effect within a limited training time. However, increasing the hyperparameter values for hard losses ($\beta_h$) does not yield such effects.
> - Additionally, we provide a theoretical analysis of the enhancement in the teacher's performance achieved by distilling knowledge from the student in **W1-1** of reviewer BAue. For more details, please see our response to W1-1 of reviewer BAue.
> - We will add this valuable discussion in the 5.2 section of this paper, highlighting the effect of increasing hyperparameter values for soft losses that can potentially shorten training time for the larger model, leading to a larger distillation effect. We hope that this result addresses your concern and the discussion strengthens our paper. We thank you so much for your helpful and insightful suggestion.
>
> ***
> **2. Using 6-layer BERT as the teacher and 4-layer BERT as the student seems like an unusual configuration. A more reasonable setting would be to use 12-layer BERT as the teacher and 6-layer or 4-layer BERT as the student model.**
>
> - Thank you for the valuable feedback from the reviewer. To address the concerns raised during the rebuttal process, we conducted experiments in a typical setting using a 12-layer BERT as the teacher model and a 6-layer BERT as the student model to demonstrate the effectiveness of the proposed CTCD approach.
> - Due to limited resources in our university environment and the constraints of the rebuttal period, we focused on validating the proposed CTCD approach during the fine-tuning stage, instead of resource-intensive pre-training, which would have required training a large model on a large text corpus from scratch. It would be greatly appreciated of your consideration of this situation.
> - Specifically, we begin with the same pre-trained 12-layer BERT as the teacher model and the same pre-trained 6-layer BERT as the student model. Then, the proposed CTCD approach involves co-training the teacher and student models, allowing them to distill knowledge from each other for 3 epochs on a target downstream task within the GLUE benchmark. In comparison, the conventional approach of One-way Knowledge Distillation (One-way KD) follows a sequential process. It starts by independently fine-tuning the teacher model for 3 epochs without any knowledge distillation (Standalone). Subsequently, knowledge is distilled from the teacher to the student over 3 epochs. We report the final performance at the last epoch as follows:
>
> |                      |          |    MNLI    |     QQP    |    QNLI    |    SST2    |    CoLA    |    STSB    |    MRPC    |     RTE    |   Average           |
> |:------------------------------:|:---------------------:|:----------:|:----------:|:----------:|:----------:|:----------:|:----------:|:----------:|:----------:|:----------:|
> |                                |                       |    Acc.    |    Acc.    |    Acc.    |    Acc.    |  Matthew.  |  Pearson.  |    Acc.    |    Acc.    |    |
> | 12-layer   BERT      (Teacher) |       Standalone      |   83.66    |   90.77    |   91.36    |   91.46    |   58.40    |   88.21    |   84.56    |   68.23    |   82.08    |
> | 12-layer   BERT      (Teacher) | **The proposed CTCD** | **84.42** | **91.04** | **91.51** | **91.97** | **59.81** | **88.70** | **85.29** | **68.95** | **82.71** |
> |  6-layer   BERT      (Student) |       One-way KD      |   81.28    |   89.12    |   88.76    |   90.48    |   52.33    |   86.28    |   84.56    |   **58.84**    |   78.96    |
> |  6-layer   BERT      (Student) | **The proposed CTCD** | **81.67** | **90.38** | **89.55** | **91.06** | **53.44** | **86.71** | **84.80** | 58.48 |   **79.51**    |
>
> - As shown in Table, when considering the teacher model, the CTCD approach demonstrates clear strengths over the Standalone method. Across tasks, CTCD consistently achieves higher accuracy and outperforms Standalone by empirically verifying the robustness of the CTCD approach in enhancing the teacher's performance through knowledge co-training. On the student model side, the comparison between One-way KD and CTCD reveals the superiority of our CTCD method. The proposed CTCD method consistently yields better accuracy and correlation metrics in all tasks except for RTE task, showing its efficacy in distilling knowledge bidirectionally between the teacher and student.
> - Therefore, we have successfully demonstrated the consistent strength of the proposed CTCD approach for both **12-layer BERT teacher and 6-layer BERT student models** in various tasks of the GLUE benchmark, confirming the bidirectional knowledge transfer effect during co-training of the models, when compared to standalone and one-way knowledge transfer approaches, across **both the pre-training and fine-tuning stages**. We believe that including the above discussions and additional experimental results in the revision significantly strengthens our paper. Thank you for the constructive comments.
>
> ***
> Once again, we extend our sincere appreciation for your time and dedication in reviewing our paper and we strongly believe that our paper be strengthened thanks to the reviewer's valuable comments.

---

### Official Review · Reviewer_Syen · 2023-08-02

**Soundness:** 4

**Excitement:**

3: Ambivalent: It has merits (e.g., it reports state-of-the-art results, the idea is nice), but there are key weaknesses (e.g., it describes incremental work), and it can significantly benefit from another round of revision. However, I won't object to accepting it if my co-reviewers champion it.

**Paper Topic And Main Contributions:**

This paper proposes a novel framework called Co-Training and Co-Distillation (CTCD) for knowledge distillation. This framework simultaneously pre-trains both the teacher and student models, with bidirectional knowledge distillation between the two. Experimental results on the GLUE benchmark demonstrate the superiority of this approach.

**Reasons To Accept:**

1. The paper is is well written and easy to understand.
2. The proposed Co-Training and Co-Distillation method is sensible, given the preliminary verification of reversed knowledge distillation.
3. The authors perform extensive ablation studies to validate the effectiveness of CTCD.

**Reasons To Reject:**

I don’t find significant flaws in this paper. There are some minor suggestions:

1. Providing implementation details for the fine-tuning phase would be beneficial.
2. It would be interesting to explore whether this framework remains effective during a fine-tuning only stage. I would expect some experiments and analysis of this scenario.

**Reproducibility:**

3: Could reproduce the results with some difficulty. The settings of parameters are underspecified or subjectively determined; the training/evaluation data are not widely available.

**Reviewer Confidence:**

4: Quite sure. I tried to check the important points carefully. It's unlikely, though conceivable, that I missed something that should affect my ratings.

---

> ### Author Rebuttal · Authors · 2023-08-28
>
> We sincerely appreciate your positive feedback that our work is **well written**, **easy to understand**, **sensible**, and **performs extensive ablation studies**. We have responded your minor suggestions as follows:
>
> ***
> **1. Providing implementation details for the fine-tuning phase would be beneficial.**
> - We thank you for your constructive comments. In the process of fine-tuning models, we employed the code provided by Hugging Face [1]. To ensure a fair comparison, we adhered to the guidelines outlined in the code for both the proposed CTCD and the one-way KD methods. Specifically, we employed key parameters, including a learning rate of 2e-5, and a training duration of 3 epochs for most tasks except for MRPC task and 5 epochs for the MRPC task within the GLUE benchmark. Following distilBERT [2], across all tasks, we maintained a consistent batch size of 32, accompanied by a maximum sequence length of 128. We will include the experimental details for the fine-tuning phase to improve the clarity in the revision. We sincerely thank you for your valuable and insightful suggestion once again.
>
> ***
> **2. It would be interesting to explore whether this framework remains effective during a fine-tuning only stage. I would expect some experiments and analysis of this scenario.**
> - As the reviewer suggested, during the rebuttal period, we validated the proposed CTCD approach during the fine-tuning stage. Specifically, we begin with the same pre-trained 12-layer BERT as the teacher model and the same pre-trained 6-layer BERT as the student model. Then, the proposed CTCD approach involves co-training the teacher and student models, allowing them to distill knowledge from each other for 3 epochs on a target downstream task within the GLUE benchmark. In comparison, the conventional approach of One-way Knowledge Distillation (One-way KD) follows a sequential process. It starts by independently fine-tuning the teacher model for 3 epochs without any knowledge distillation (Standalone). Subsequently, knowledge is distilled from the teacher to the student over 3 epochs. We report the final performance at the last epoch as follows:
>
> |                      |          |    MNLI    |     QQP    |    QNLI    |    SST2    |    CoLA    |    STSB    |    MRPC    |     RTE    |   Average           |
> |:------------------------------:|:---------------------:|:----------:|:----------:|:----------:|:----------:|:----------:|:----------:|:----------:|:----------:|:----------:|
> |                                |                       |    Acc.    |    Acc.    |    Acc.    |    Acc.    |  Matthew.  |  Pearson.  |    Acc.    |    Acc.    |    |
> | 12-layer   BERT      (Teacher) |       Standalone      |   83.66    |   90.77    |   91.36    |   91.46    |   58.40    |   88.21    |   84.56    |   68.23    |   82.08    |
> | 12-layer   BERT      (Teacher) | **The proposed CTCD** | **84.42** | **91.04** | **91.51** | **91.97** | **59.81** | **88.70** | **85.29** | **68.95** | **82.71** |
> |  6-layer   BERT      (Student) |       One-way KD      |   81.28    |   89.12    |   88.76    |   90.48    |   52.33    |   86.28    |   84.56    |   **58.84**    |   78.96    |
> |  6-layer   BERT      (Student) | **The proposed CTCD** | **81.67** | **90.38** | **89.55** | **91.06** | **53.44** | **86.71** | **84.80** | 58.48 |   **79.51**    |
>
> - As shown in Table, when considering the teacher model, the CTCD approach demonstrates clear strengths over the Standalone method. Across tasks, CTCD consistently achieves higher accuracy and outperforms Standalone by empirically verifying the robustness of the CTCD approach in enhancing the teacher's performance through knowledge co-training. On the student model side, the comparison between One-way KD and CTCD reveals the superiority of our CTCD method. The proposed CTCD method consistently yields better accuracy and correlation metrics in all tasks except for RTE task, showing its efficacy in distilling knowledge bidirectionally between the teacher and student.
> - Therefore, we have successfully demonstrated the consistent strength of the proposed CTCD approach for both teacher and student models in various tasks of the GLUE benchmark, confirming the bidirectional knowledge transfer effect during co-training of the models, when compared to standalone and one-way knowledge transfer approaches, across both the pre-training and fine-tuning stages. We believe that including the above discussions and additional experimental results in the revision significantly strengthens our paper. Thank you for the constructive comments.
>
> ***
> [1] https://github.com/huggingface/transformers/tree/main/examples/pytorch/text-classification
>
> [2] Sanh, Victor, et al. "DistilBERT, a distilled version of BERT: smaller, faster, cheaper and lighter." arXiv 201
>
> ***
> Thank you for your helpful comments. We believe that our paper will become much stronger after including the additional experiments you have suggested.

---

### Official Review · Reviewer_BAue · 2023-08-05

**Soundness:** 3

**Excitement:**

3: Ambivalent: It has merits (e.g., it reports state-of-the-art results, the idea is nice), but there are key weaknesses (e.g., it describes incremental work), and it can significantly benefit from another round of revision. However, I won't object to accepting it if my co-reviewers champion it.

**Paper Topic And Main Contributions:**

The paper proposes a novel knowledge distillation framework called Co-Training and Co-Distillation (CTCD) to improve model performance while compressing large pre-trained language models (PLMs). The key idea is to co-train a large teacher model and a small student model together, enabling bidirectional knowledge transfer between them. The results on GLUE are reported. Overall, I think the idea of mutual knowledge transfer is interesting but the actual performance gains seem quite small based on the experiments.

**Questions For The Authors:**

1. what does "training from scratch" mean for co-training? Are the student and teacher not initialized from pre-trained BERTs? The wording is misleading and not compatible with the figures.

**Reasons To Accept:**

1. The idea of bidirectional knowledge transfer between teacher and student during co-training is novel.
2. The experiments show distilling knowledge from the student to teacher improves the teacher's performance, which in turn benefits the student.
3. The student model compressed by CTCD outperforms the standalone trained original teacher model on GLUE.

**Reasons To Reject:**

1. The paper studies the case where the student distills knowledge to the teacher, which improves the teacher's performance. However, the improvements could potentially be due to regularization effects rather than distillation as claimed, since all fine-tuning is performed for 10 epochs and without early-stopping. The fine-tuning on GLUE without validation early-stopping usually has very high variances, proper ablation studies are needed to verify.

2. The performance gains (especially, the Community KD) over standard one-way distillation seem quite marginal based on the experiments when compared to other BERT distillation techniques like BERT-PKD, TinyBERT, MobileBERT, or BERT-of-Theseus. This is not a good signal given the training process is more complex with co-training and co-distillation compared to other distillation techniques.

3. Evaluations are limited to BERT models only. Testing on other PLMs would be more convincing.

**Reproducibility:**

3: Could reproduce the results with some difficulty. The settings of parameters are underspecified or subjectively determined; the training/evaluation data are not widely available.

**Reviewer Confidence:**

4: Quite sure. I tried to check the important points carefully. It's unlikely, though conceivable, that I missed something that should affect my ratings.

---

> ### Author Rebuttal · Authors · 2023-08-28
>
> We thank you for your constructive comments and for your positive feedback that **our idea is novel** and our findings are a reason to accept. We provide responses to your comments below:
>
> ***
> **W1-1. The teacher’s performance improvement via the reverse KD (student —> teacher) could potentially be due to regularization effects rather than distillation.**
> - We respectfully disagree with the reviewer's comment. **KD is fundamentally a learned Label Smoothing Regularization (LSR) [1, 2, 3]**. Therefore, the regularization effect of KD should **not** be a concern; instead, it serves as **theoretical evidence** that clearly **supports** the effectiveness of our method.
> - Specifically, following theoretical analysis [1],
>   - In Label Smoothing Regularization (LSR), the goal is to minimize the cross-entropy between the smoothed label distribution $q'(k)$ and the model output $p(k)$. The smoothed label distribution $q'(k)$ is created by modifying the original label distribution:
> $$q'(k)=(1-\alpha)q(k) + \alpha u(k) \tag{1}$$ which is a mixture of the original distribution $q(k)$ and a fixed distribution $u(k)$, with weight $\alpha$, Usually, the $u(k)$ is uniform distribution $u(k)= 1/K$ where $K$ is the number of classes.
>   - The cross-entropy loss $H(q',p)$ over the smoothed labels is defined as follows:
> $$H(q',p) =-\sum_{k=1}^K q'(k) \log p(k) =(1-\alpha)H(q,p) + \alpha H(u, p) =(1-\alpha)H(q,p) + \alpha (D_{KL}(u, p) + H(u)) \tag{2}$$
> where $D_{KL}$ is the Kullback-Leibler divergence and $H(u)$ is a constant for the fixed uniform distribution $u(k)$. Thus, the loss function of label smoothing is denoted as
> $$L_{LS} = (1 - \alpha) H(q,p) + \alpha D_{KL} (u, p) \tag{3}$$
>   - On the other hand, the goal of KD is to mimic teacher behavior by minimizing the cross-entropy loss (hard loss) and KL divergence between the predictions of student and teacher as
> $$L_{KD} = (1 - \alpha) H(q,p) + \alpha D_{KL} (p^t_\tau, p_\tau) \tag{4}$$
>   - Upon comparing Eq. (3) and Eq. (4), the sole distinction arises from the fact that $p^t_\tau$ in $D_{KL} (p^t_\tau, p_\tau)$ represents a distribution derived from a teacher model, while $u(k)$ in $D_{KL}(u, p)$ signifies a pre-defined uniform distribution. In this context, we can regard KD as a special case of LSR where the smoothing distribution is acquired through learning rather than being predefined. This notion is further supported by regarding the regularization term $D_{KL}(u, p)$ as a virtual teacher model in KD, which assigns a uniform probability to all classes.
>   - Since $D_{KL} (p^t_\tau, p_\tau)=H(p^t_\tau, p_\tau) - H(p^t_\tau)$, where the entropy $H(p^t_\tau)$ is constant, we can reformulate Eq. (4) as
> $$L_{KD} = (1-\alpha)H(q,p) + \alpha H(p^t_\tau, p_\tau)$$
> If we set the temperature $\tau=1$, we have $L_{KD}=H(\tilde{q}^t, p)$, where $\tilde{q}^t$ is
> $$\tilde{q}^t=(1-\alpha)q(k)+\alpha p^t(k) \tag{5}$$
>   - By comparing Eq. (5) and Eq. (1), we can see more clearly that KD is a specific instance of LSR, utilizing the learned distribution $p^t(k)$ from a trained teacher rather than a uniform distribution.
> - To address the concerns from reviewer BAue's comments, we will include a discussion of the regularization effects through the distillation as the theoretical analysis to support the effectiveness of our proposed method in the revision. We appreciate the insightful comments and this discussion strengthens our paper.
>
> **W1-2. All fine-tuning is performed for 10 epochs and without early-stopping. The fine-tuning on GLUE without validation early-stopping usually has very high variances.**
>  - This is a **factual misunderstanding**. As clearly mentioned in **Lines 111-120** and also depicted in **Figure 1**, our approach focused on validating the effectiveness of the proposed CTCD method in the **task-agnostic (pre-training) phase, not the task-specific (fine-tuning) phase**. In other words, in the **pre-training** phase, we conducted training for **10 epochs** (up to 20 epochs) and performed knowledge distillation. During the **fine-tuning** phase, following Hugging Face's guidelines [4], we **applied early stopping** by training all baselines, including the proposed method, for only 3 epochs (5 epochs for MRPC). Therefore, we strongly believe that our proposed method is **not** subject to the higher variance issue caused by longer training in the fine-tuning phase.
> - Nonetheless, to address the reviewer’s concern, we conducted experiments to apply our CTCD approach at the fine-tuning phase to verify the efficacy of our method. Please refer to our response for **W3**.
> - We hope that this discussion clears up the confusion of the reviewer regarding the training length of the proposed method. In the revision, we will add experimental details regarding fine-tuning and provide a clear distinction between fine-tuning and pre-training. Thanks to the discussion with the reviewer, we believe that the clarity of the paper has been improved. We appreciate the reviewer's constructive comments.
>
> ***
> **W2. The performance gains over standard one-way distillation seem quite marginal based on the experiments when compared to other BERT distillation techniques like BERT-PKD, TinyBERT, MobileBERT, or BERT-of-Theseus.**
> - Directly comparing the proposed methods with BERT-PKD, TinyBERT, MobileBERT, and BERT-of-Theseus merely based on performance would be **unfair** since they have employed **extra techniques** beyond the KD algorithm for performance improvement. For example, TinyBERT uses data augmentation, MobileBERT focuses on student architecture design, BERT-of-Theseus applies progressive module replacement, and BERT-PKD conducts KD from multiple intermediate layers. **All of these extra techniques can be integrated into our CTCD algorithm orthogonally**, by replacing one-way KD in their frameworks using our CTCD algorithm instead of directly comparing them with it, thereby potentially generating additional performance improvement or synergistic effects easily.
> - Our goal is **not achieving SOTA performance, but rather verifying whether the knowledge distillation algorithm can contribute not only to model compression but also to performance enhancement**. Therefore, to clearly validate the effectiveness of our proposed method, we conducted a performance comparison between one-way KD and the proposed CTCD method in a controlled environment without employing extra techniques like architecture designs or data augmentation. Through extensive ablation studies and rigorous comparisons, we successfully demonstrated two novel findings: 1)  during co-training, distilling knowledge from the student to the teacher improves the teacher's performance and 2) the performance gains of the teacher model, in turn, benefits the student.
> - We believe that integrating our CTCD algorithm with other extra techniques is a promising and interesting research direction. We will include it as future work in the discussion section of the revision. We sincerely appreciate the reviewer’s constructive comments.
>
> ***
>
> **W3. Evaluations are limited to BERT models only. Testing on other PLMs would be more convincing.**
> - As our approach has focused on the task-agnostic phase that involves training large language models from scratch on a large text corpus during the pre-training stage, training language models that are larger than BERT is challenging with limited resources from our university environment and the constraints of the rebuttal period. It would be greatly appreciated of your consideration of this situation.
> - However, we believe that the proposed algorithm is convincingly effective. As acknowledged by the **Reviewer Syen**, our extensive ablation study has supported the effectiveness of the proposed CTCD. Furthermore, by replacing solely the conventional one-way KD algorithm with the proposed CTCD algorithm at the pre-training stage **without any extra techniques**, we achieved consistent performance improvements across various downstream tasks in the GLUE benchmark on average, we attained a significant performance boost of 1.66. To see the clear efficacy of the proposed algorithm, we did not use any supplementary techniques like additional KD at the fine-tuning stage or data augmentation.
> - Furthermore, during the rebuttal period, to provide additional convincing experimental results, we conducted experiments to demonstrate the efficacy of CTCD at the **fine-tuning stage** as well as the pre-training stage. Specifically, we begin with the same pre-trained 12-layer BERT as the teacher model and the same pre-trained 6-layer BERT as the student model. Then, the proposed CTCD approach involves co-training the teacher and student models, allowing them to distill knowledge from each other for 3 epochs on a target downstream task within the GLUE benchmark. In comparison, the conventional approach of One-way Knowledge Distillation (One-way KD) follows a sequential process. It starts by independently fine-tuning the teacher model for 3 epochs without any knowledge distillation (Standalone). Subsequently, knowledge is distilled from the teacher to the student over 3 epochs. We report the final performance at the last epoch as follows:
>
> |                      |          |    MNLI    |     QQP    |    QNLI    |    SST2    |    CoLA    |    STSB    |    MRPC    |     RTE    |   Average           |
> |:------------------------------:|:---------------------:|:----------:|:----------:|:----------:|:----------:|:----------:|:----------:|:----------:|:----------:|:----------:|
> |                                |                       |    Acc.    |    Acc.    |    Acc.    |    Acc.    |  Matthew.  |  Pearson.  |    Acc.    |    Acc.    |    |
> | 12-layer   BERT      (Teacher) |       Standalone      |   83.66    |   90.77    |   91.36    |   91.46    |   58.40    |   88.21    |   84.56    |   68.23    |   82.08    |
> | 12-layer   BERT      (Teacher) | **The proposed CTCD** | **84.42** | **91.04** | **91.51** | **91.97** | **59.81** | **88.70** | **85.29** | **68.95** | **82.71** |
> |  6-layer   BERT      (Student) |       One-way KD      |   81.28    |   89.12    |   88.76    |   90.48    |   52.33    |   86.28    |   84.56    |   **58.84**    |   78.96    |
> |  6-layer   BERT      (Student) | **The proposed CTCD** | **81.67** | **90.38** | **89.55** | **91.06** | **53.44** | **86.71** | **84.80** | 58.48 |   **79.51**    |
>
> - As shown in Table, when considering the **teacher model**, the CTCD approach demonstrates clear strengths over the Standalone method. Across tasks, **CTCD consistently achieves higher accuracy and outperforms Standalone** by empirically verifying the efficacy of the CTCD approach in enhancing the teacher's performance through knowledge co-training. On the **student model** side, the comparison between One-way KD and CTCD reveals the superiority of our CTCD method. The proposed **CTCD method consistently yields better accuracy and correlation metrics** in all tasks except for RTE task, showing its efficacy in distilling knowledge bidirectionally between the teacher and student.
> - Therefore, we have successfully demonstrated the consistent strength of the proposed CTCD approach for both teacher and student models in various tasks of the GLUE benchmark, confirming the bidirectional knowledge transfer effect during co-training of the models, when compared to standalone and one-way knowledge transfer approaches, across **both the pre-training and fine-tuning stages**. We believe that including the above discussions and additional experimental results in the revision significantly strengthens our paper. Thank you for the constructive comments.
>
> **Q1. What does "training from scratch" mean for co-training? Are the student and teacher not initialized from pre-trained BERTs? The wording is misleading and not compatible with the figures.**
> - As we explained in **W1-2** of our response, we have applied our CTCD approach at pre-training phase (not fine-tuning phase), thus, the student and teacher are not initialized from pre-trained BERTs and they are co-trained from scratch. Therefore, the wording is correct and compatible with the figures. We will clarify this by emphasizing the pre-training phase in the revision. Thank you for your helpful feedback.
>
> [1] Yuan et al., Revisiting knowledge distillation via label smoothing regularization, CVPR 2020.
>
> [2] Tang, Jiaxi, et al., Understanding and improving knowledge distillation, arXiv 2020.
>
> [3] Kim, Taehyeon, et al., Comparing Kullback-Leibler Divergence and Mean Squared Error Loss in Knowledge Distillation, IJCAI 2021
>
> [4] https://github.com/huggingface/transformers/tree/main/examples/pytorch/text-classification
>
> ***
> We believe that including the above discussions and additional experimental results significantly strengthens our paper. Thank you for the constructive comments.

---

### Meta-Review · Area_Chair_yAnf · 2023-09-18

**Recommendation:** 4

**Metareview:**

The paper proposes a novel distillation framework where student and teacher models are co-trained to enable superior transfer between them.  The framework is novel and the paper is well written. The rebuttal has acknowledged some of the limitations around the experimental setting (instantiations of the student and teacher models). The performance gains are small but significant. I think the community would benefit from this novel framework.

---

### Decision · Program_Chairs · 2023-10-07

**Decision:**

Accept-Findings

**Comment:**

The paper proposes a novel distillation framework where student and teacher models are co-trained to enable superior transfer between them.  The framework is novel and the paper is well written. The rebuttal has acknowledged some of the limitations around the experimental setting (instantiations of the student and teacher models). The performance gains are small but significant. I think the community would benefit from this novel framework.